# Prognostic nomogram for uncontrolled type 2 diabetes using Thailand nation-wide cross-sectional studies

**Teeraboon Lertwanichwattana**[1], **Picha Suwannahitatorn**[2], **Mathirut Mungthin**[2], **Ram Rangsin**[1]*

**1** Department of Military and Community Medicine, Phramongkutklao College of Medicine, Bangkok, Thailand, **2** Department of Parasitology, Phramongkutklao College of Medicine, Bangkok, Thailand

* rrangsin@pmk.ac.th

## Abstract

### Background

Uncontrolled type 2 diabetes (T2DM) and limited hemoglobin A1c (HbA1c) levels examination are a burden in community hospitals in Thailand. The nomogram from the patients' information might be a practical solution to identify a high-risk group of diabetic complications. Thus, this study aimed to establish an effective prognostic nomogram for patients with uncontrolled T2DM.

### Methods

Sequential nationwide cross-sectional studies of T2DM patients in 2018 and 2015 were utilized for development and validation groups, respectively, with this chronological order aiming to capture recent trends during development and assess the nomogram's robustness across diverse timeframes. The predictive outcome was uncontrolled T2DM, defined as HbA1c $\geq$9%. The model was determined by multivariable regression analysis and established an effective prognostic nomogram. The receiver operating characteristic curve, Hosmer-Lemeshow goodness of fit test, and decision curve analysis (DCA) was applied to evaluate the performance of the nomogram.

### Results

In 2018, 24% of the 38,568 participants in the development group had uncontrolled T2DM (defined as Hba1c $\geq$9%). The predictive nomogram of uncontrolled diabetes consisted of demographic characteristics, prescription medications, history of diabetic complications, and laboratory results (C-statistic of 0.77). The goodness of fit test and DCA showed good agreement between the result and clinical application for T2DM.

### Conclusion

The predictive nomogram demonstrates simplicity, accuracy, and valuable prediction to enhance diabetic care in resource-limited countries, including Thailand.

**Data Availability Statement:** The data underlying this study are third party data owned by Thai MedResNet (Thailand Medical Research Network).

These data are available upon request pending Thai MedResNet approval from Data Archive for Maximal Utilization System (DAMUS; www.damus. in.th/damus). To seek access these data, qualified interested researchers may contact: Channarong Chokbumrungsuk Data Management Unit (DMU) manager Medical Research Network of the Consortium of Thai Medical Schools (MedResNet Thailand) 196 Phaholyothin Rd, Lad Yao, Chatuchak, Bangkok, 10900 Phone: +66 2940 5181 Fax: +66 2940 5184 Email: damus@thaimedresnet.org, channarong@thaimedresnet.org.

**Funding:** The authors did not receive any third-party support in conducting this research, analyzing the data, specific funding, or preparing the manuscript for submission.

**Competing interests:** The authors have declared that no competing interests exist.

## Background

Type 2 diabetes mellitus (T2DM) is a chronic disease constituting one of four priority non-communicable diseases according to the World Health Organization (WHO) and targeted for action by world leaders. The most common is type 2 diabetes resulting from the body's ineffective use of insulin. The International Diabetes Federation reports that approximately 463 million adults (20 to 79 years) were living with diabetes in 2019 and projects that this number will increase to 700 million by 2045. Thailand is among the top five countries with the highest number of people with diabetes (20 to 79 years), 4.30 million in 2019, and the rate of undiagnosed diabetes reached nearly 50% [1]. Another study conducted in Thailand using the 5th Thai National Health Examination Survey found that the age-adjusted prevalence increased from 7.70% in 2004 to 7.80% in 2009 and 9.90% in 2014. In addition, after focusing on diabetes-related and related causes of death, we discovered that all 3.70 million deaths were due to high blood glucose levels, including 1.50 million deaths from diabetes and 2.20 million deaths from cardiovascular and other illnesses caused by uncontrolled optimal glucose levels. This reflection represents mortality resulting from high blood glucose, causing a significant mortality burden more than diabetes [2].

Complications of T2DM include cardiovascular diseases (atherosclerotic cardiovascular disease (ASCVD) and heart failure) and microvascular complications (chronic kidney diseases (CKD), diabetic retinopathy (DR), and neuropathy), according to the American Diabetes Association (ADA) [3, 4]. Diabetes was associated with a two- to fourfold increased incidence of atherosclerotic cardiovascular disease compared to those without diabetes [5]. The cost of treatment following diabetes complication was as high as $11,664 for acute myocardial infarction, $11,635 for acute ischemic stroke, and $37,611 for coronary artery bypass grafting [6].

Glycemic control is assessed by the HbA1c measurement, continuous glucose monitoring (CGM), and self-monitoring of blood glucose (SMBG). First, HbA1c reflects the average blood glucose level over approximately three months. The performance of the test must be standardized by the National Glycohemoglobin Standardization Program (NGSP)-certified assays [7]. The optimal HbA1c level will likely decrease both macrovascular and microvascular diabetes complications, particularly in the normal range (<6.00%) [8]. Lastly, CGM and SMBG only play an essential role in self-awareness, medication adjustment, and prevention of hypoglycemia. In Thailand, Sakboonyarat B. et al. reported that the proportion of adequate glycemic control (HbA1c level <7.00%) among patients with T2DM under continuous care in public hospitals was 35.60% (95%CI; 35.00–36.20%) in 2018 [9].

The American Association of Clinical Endocrinologists (AACE) and the American Diabetes Association (ADA) recommended that insulin administration be strongly considered for T2DM, with HbA1c levels exceeding 9.00% [10] and 10% [11], respectively. If the HbA1c level is greater than 9.00%, earlier insulin therapy initiation should be recommended to enhance glucose control, which could result in cardiovascular benefits for hypertensive patients [12].

Assessing glycemic control using HbA1c, according to ADA 2020, recommends evaluating the level quarterly per year among patients whose therapy has changed or who are not meeting glycemic goals [10]. Moreover, AACE 2020 recommends assessing every three months, and when the patient does not achieve the goal in three months, proceed to the next level of therapy instantly [11]. Thailand's tight control of diabetes recommends assessing glycemic control using HbA1c every three to six months or at least once yearly [8]. In Thailand, only 76.40% (75.90–76.90) of patients with T2DM diabetes received an HbA1c evaluation at least once yearly in 2018 [9]. The coverage of having at least one annual HbA1c test among patients with T2DM varied by health region in Thailand. The lowest coverage was found in Health Region 4 (65.60%) [13]. One of the main reasons the HbA1c test could not be thoroughly provided is

the limited availability of certified HbA1c tests, which cannot be performed primarily in their laboratories in many public hospitals in Thailand.

The Diabetes Control and Complications Trial and the United Kingdom Prospective Diabetes Study from the NGSP found a direct relationship between HbA1c levels and diabetic outcome risks, indicating the necessity to standardize the hbA1c assessment [7]. Some laboratories in limited resources countries do not evaluate the HbA1c quality assessment using NGSP. Furthermore, screening models for practical prognostic uncontrolled diabetes would be helpful. A nomogram for uncontrolled diabetes, constituting a graphical prediction model, could benefit diabetic care where the HbA1c test was considered scarce.

Moreover, due to the limited budget distributed by the Centre of Healthcare in Thailand, which spent the monthly cost per head of patient based on the International Classification of Diseases, 10th Revision (ICD-10), regardless of whether a patient was able to control his or her disease well or not, no patient was able to control his or her disease adequately. Consequently, community hospitals were constrained by a limited budget and were required to use it effectively.

Therefore, this study aimed to develop a nomogram model for predicting uncontrolled diabetes (defined as HbA1c 9%) based on individual baseline characteristics, medications taken, disease complications, and laboratory results routinely available from public datasets. This model would primarily aid in identifying patients at risk for uncontrolled diabetes and in ensuring that these patients undergo more frequent HbA1c testing.

## Methods

### Study design

The database from the study, "National trends in the prevalence of glycemic control among patients with type 2 diabetes receiving continuous care in Thailand from 2011 to 2018 (Thailand DM/HT studies)" [9], was used in this study. This fully anonymized database is publicly available at the National Health Security Office (NHSO). The studies comprised a series of annual cross-sectional studies from 2011 to 2015 and 2018 to evaluate the status of diabetes care among patients with type 2 diabetes attending public hospitals of the Ministry of Public Health nation-wide in Thailand. Model prediction and validation were conducted from the database in 2018 and 2015. This study was reviewed and approved by the Royal Thai Army Medical Department Institutional Review Board, reference number S043h/64.

### Subjects

In all, 38,568 and 32,616 participants, identified from the 2018 and 2015 Thailand DM /HT studies, were used as the development and validation groups, respectively. These participants comprised Thai citizens receiving a diagnosis of T2DM and receiving continuous care in government hospitals nation-wide for more than one year. All Thai citizens can access the health care system due to health care coverage. This depends on the hospital's jurisdiction, residential area, and social work. The study participants from development and validation groups included all three significant schemes consisting of universal coverage (UC), civil servant medical benefit scheme (CSMBS), and social security scheme (SSS). The inclusion criteria included patients with T2DM aged at least 18 years and having been treated for diabetes in their hospital for more than one year. All patients who had participated in a clinical trial were excluded.

### Data collection

The Thailand DM/HT study database in 2015 and 2018 was retrieved from NHSO Central Data Management Unit. Details of the Thailand DM/HT study have been reported elsewhere

[9]. The data were collected from 987 hospitals in Thailand, including 151 standard/advanced level hospitals and 836 community-level hospitals. Patients diagnosed with T2DM were randomly selected and registered at each hospital. A standardized case report form was used to obtain the required information from the medical records for diabetes care, including demographic data, treatments, laboratory tests, and clinical outcomes. The case report forms of the enrolled participants were sent to the Thailand DM/HT study Data Management Unit. The final database was kept at the NHSO Data Management Unit. To ensure the privacy and confidentiality of the subjects involved, we took careful measured to remove any personally identifiable information from the data before accessing it. Our study solely focused on analyzing aggregated patterns and developing predictive models using the data. We ensured that no information capable of tracking or identifying individual subjects was included in our analysis or reported results.

## Measurements

The obtained participant information from the database included socio-demographic information (gender, age, occupation, religion, health insurance), weight, height, body mass index (BMI), waist circumference, smoking habits (current smokers, former smokers, and non-smokers), alcohol consumption (current drinkers, former drinkers, and non-drinkers), hospitalization, blood chemistry data including fasting plasma glucose (FPG), HbA1c, serum creatinine, history of use of antihyperglycemics, antihypertensives and lipid-lowering medications and glomerular filtration rate calculated according to the epidemiologic collaboration formula. The recorded database obtained from the ICD-10 code-based definition was used to identify hypertension (I10-I15), dyslipidemia (E78), gout (M10), renal insufficiency (N18), diabetic kidney diseases (E11.2), diabetic retinopathy (E11.3), cerebrovascular accident (I60-I69), and coronary artery disease (I20-I25).

## Statistical analysis

The present study's primary outcome was HbA1c $\geq$9%. Categorical variables were classified by clinical findings, which were expressed as frequency and percentage. Continuous variables were expressed as mean and standard deviation. Descriptive statistics were used to analyze the difference between the development and the validation groups. The Chi-square test was used to compare the frequency distribution of categorical variables. The Shapiro-Wilk test and independent sample t-test were used for continuous variables of the normal distribution test and the difference between groups, respectively.

All the variables were evaluated using univariable logistic regression analysis. The model was initiated with 26 candidate variables and reduced to find the best-fitting model. Multivariable stepwise (backward) logistic regression analysis was conducted to assess factors associated with uncontrolled diabetes, using a significance level of $p$-value $< 0.2$ as the criterion to remove variables from the models. Odds ratios (OR) and 95% confidence intervals (95% CI) were calculated for significant variables. The Variance Inflation Factor (VIF) was employed to address multicollinearity among independent variables. Variables with statistical significance in both univariable and multivariable analyses were combined to predict uncontrolled diabetes.

The final prediction model was applied to develop an effective prognostic nomogram. The nomogram performance was evaluated regarding differentiation, calibration, and clinical validity. Receiver Operating Characteristic (ROC) curve analysis and the area under the curve (AUC) were used to measure nomogram accuracy. The AUC neared 1, the more the maximum AUC, and the perfection in the differentiation between the diseased and non-diseased

improved. The true positive fraction (sensitivity), false-positive fraction (1-specificity), positive predictive value, negative predictive value, positive likelihood ratio, and negative likelihood ratio were established. Youden's index maximizes the difference between TPF (sensitivity) and FPF (1-specificity), which calculates the optimal cut-off point on the ROC curve. The model correction used the Hosmer-Lemeshow goodness-of-fit test with a *p*-value greater than 0.05. No distinct difference was found between the predicted and actual values. Decision curve analysis was used to evaluate the clinical validity of the model. In addition, the ROC curve point defined the optimal cut-off value, which was selected using Youden's index (sensitivity+-specificity-1).

## Results

### Demographic characteristics of the study participants

In 2018 and 2015, respectively, 38,568 and 32,616 T2DM patients were recruited for the development and validation groups. The characteristics of the enrolled patients are presented in Table 1. Nearly two-thirds of enrolled patients in 2018 and 2015 were female (66.95% and 67.49%, respectively). The mean age of enrolled patients in 2018 and 2015 was 62.34 ± 11.00 years and 61.50 ± 10.96 years, respectively. Most study participants (78.48%, and 76.36% in 2018 and 2015, respectively) were under the UC scheme. Approximately 70% (73.49% and 67.02% in 2018 and 2015, respectively) of study participants received care in community hospitals. Three of four study participants presented co-morbidity and high blood pressure (78.08% and 78.24% in 2018 and 2015, respectively). Approximately one-quarter of patients with T2DM (24.00%, and 25.28% in 2018 and 2015, respectively) were classified as having an uncontrolled diabetic status (HbA1c ≥9%).

### Predictive variables

The final prediction model to identify the independent factors associated with uncontrolled diabetes (HbA1c ≥9%) was analyzed using univariable and multivariable stepwise (backward) logistic regression analysis (Tables 2 and 3). The final prediction model included 13 variables associated with uncontrolled diabetes. Being a female (AOR 1.25, 95%CI 1.17–1.34), older age (AOR 0.35, 95%CI 0.27–0.45), CSMBS (AOR 0.90, 95%CI 0.83–0.99), SSS (AOR 0.84, 95%CI 0.72–0.99), receiving care from community hospitals (AOR 1.22, 95%CI 1.08–1.38), residing in the central region (AOR 0.83, 95%CI 0.76–0.90), residing in the northeastern region (AOR 1.41, 95%CI 1.29–1.53), residing in the southern region (AOR 1.34, 95%CI 1.22–1.47), duration of diabetes more than ten years (AOR 1.24, 95%CI 1.16–1.33), BMI (AOR 0.99, 95%CI 0.98–1.00), higher FPG level (AOR 9.65, 95%CI 7.70–12.09), biguanide prescription (AOR 1.47, 95%CI 1.36–1.60), sulfonylurea prescription (AOR 1.66, 95%CI 1.24–1.49), thiazolidinedione prescription (AOR 1.36, 95%CI 1.24–1.49), insulin prescription (AOR 4.31, 95%CI 3.99–4.66), and diagnosis of DR (AOR 1.19, 95%CI 1.04–1.36) were considered as independent predictive variables for uncontrolled diabetes. In addition, multi-collinearity was investigated using the variance inflation factor of each variable, for which the maximum variance inflation factor was 1.36, and the minimum was 1.01, indicating no multi-collinearity.

### Development and validation of uncontrolled diabetes predicting model

Based on the multivariable stepwise (backward) logistic regression model, a nomogram (Fig 1) was constructed to predict uncontrolled diabetes. The lowest line of the nomogram, labeled "Total score," was used to calculate the score associated with each of the 13 variables. The top 13 lines ("Sulfonylurea" through "Thiazolidinediones") were the associated factors used in the

**Table 1. Comparison of demographic characteristics between the development group and validation group.**

| Characteristics | Development Group | Validation Group | P-value |
|---|---|---|---|
| | 2018 | 2015 | |
| | N = 38,568 | N = 32,616 | |
| | n (%) | n (%) | |
| **Gender** | | | 0.123 |
| Male | 12,748 (33.05) | 10,603 (32.51) | |
| Female | 25,820 (66.95) | 22,013 (67.49) | |
| **Age (years)** | | | |
| Mean ± SD | 62.34 ± 11.00 | 61.50 ± 10.96 | <0.001 |
| <40 | 786 (2.04) | 777 (2.38) | <0.001 |
| 40–49 | 3,874 (10.04) | 3,683 (11.29) | |
| 50–59 | 10,645 (27.60) | 9,420 (28.88) | |
| 60–69 | 13,334 (34.57) | 10,965 (33.62) | |
| ≥70 | 9,929 (25.74) | 7,771 (23.83) | |
| **Health insurance scheme** | | | <0.001 |
| UC | 30,269 (78.48) | 24,905 (76.36) | |
| CSMBS | 6,270 (16.26) | 5,716 (17.53) | |
| SSS | 1,523 (3.95) | 1,335 (4.09) | |
| Others | 506 (1.31) | 660 (2.02) | |
| **Regions** | | | <0.001 |
| North | 8,920 (23.13) | 5,133 (15.74) | |
| Central | 12,505 (32.42) | 13,410 (41.11) | |
| Northeast | 10,610 (27.51) | 9,444 (28.96) | |
| South | 6,533 (16.94) | 4,629 (14.19) | |
| **Hospital level** | | | <0.001 |
| Regional Hospitals | 2,670 (6.92) | 2,919 (8.95) | |
| General Hospitals | 7,554 (19.59) | 7,838 (24.03) | |
| Community Hospitals | 28,344 (73.49) | 21,859 (67.02) | |
| **Hypertension** | | | 0.593 |
| No | 8,455 (21.92) | 7,096 (21.76) | |
| Yes | 30,113 (78.08) | 25,520 (78.24) | |
| **Biguanide** | | | <0.001 |
| No | 9,396 (24.36) | 9,084 (27.85) | |
| Yes | 29,172 (75.64) | 23,532 (72.15) | |
| **Sulfonylurea** | | | <0.001 |
| No | 15,800 (40.97) | 12,195 (37.39) | |
| Yes | 22,768 (59.03) | 20,421 (62.61) | |
| **Thiazolidinediones** | | | <0.001 |
| No | 33,865 (87.81) | 29,269 (89.74) | |
| Yes | 4,703 (12.19) | 3,347 (10.26) | |
| **Insulin** | | | 0.032 |
| No | 30,109 (78.07) | 25,243 (77.39) | |
| Yes | 8,459 (21.93) | 7,373 (22.61) | |
| **Diabetic retinopathy** | | | <0.001 |
| No | 36,624 (94.96) | 30,573 (93.74) | |
| Yes | 1,943 (5.04) | 2,043 (6.26) | |
| **FPG (mg/dL)** | | | |
| Mean ± SD | 153.53 ± 54.29 | 153.89 ± 55.76 | 0.400 |

(*Continued*)

**Table 1.** (Continued)

| Characteristics | Development Group | Validation Group | P-value |
|---|---|---|---|
| | 2018 | 2015 | |
| | N = 38,568 | N = 32,616 | |
| | n (%) | n (%) | |
| **HbA1c (%)** | | | <0.001 |
| <9 | 25,363 (76.00) | 19,682 (74.72) | |
| ≥9 | 8,010 (24.00) | 6,659 (25.28) | |

SD: standard deviation, UC: universal coverage, SSS: social security scheme, CSMBS: civil servant medical benefit scheme, FPG: fasting plasma glucose, HbA1c: hemoglobin A1c, mg/dL: milligrams per deciliter

prediction model. The corresponding "diagnostic possibility" of "total score" was the predicted probability of uncontrolled diabetes by nomogram. The validation of the nomogram consisted of discrimination, calibration, and clinical validity test. The form of an index's receiver-operating characteristic (ROC) curves (Fig 2) of the discrimination process was implied. The area under curve (AUC) of the development and validation groups was 0.71 (95%CI 0.70–0.72) and 0.70 (95% CI 0.70–0.71), respectively. Table 4 shows the prediction performance of the nomogram in the development and validation groups. The optimum cut-off value by Youden's index was 0.23 in both the development and validation groups. The sensitivity was 64.80% and 69.70%, and the specificity was 77.00% and 70.90% in the development and validation groups, respectively. The calibration of the nomogram was evaluated by the Hosmer-Lemeshow goodness of fit test of 0.53, which was the better-predicted ability of the model, $p$-value >0.05.

## Decision curve analysis

Decision curve analysis was used in the clinical validation of uncontrolled diabetes. The green line represented the model, the blue line represented the net benefit when all participants had uncontrolled diabetes, and the red line represented the net benefit when no participants had uncontrolled diabetes. The green line model above the red horizontal line evaluated the conclusion of the decision curve. Moreover, the blue left oblique line. Thus, this model was a good assessment tool for the selection probability threshold (Fig 3).

## Discussion

Diabetes mellitus is one of the significant health problems in Thailand. The prevalence of diabetes in Thailand has increased dramatically over the decades. The National Health Examination Survey (NHES) demonstrated the increasing age-standardized prevalence of type 2 diabetes (age ≥20 years old) by history and FPG level, which were 2.30, 4.60, 6.80, 6.90, and 9.90% in 1991, 1997, 2005, 2009, and 2014, respectively [14–16]. Recently, in NHES 2020, the prevalence of type 2 diabetes (age ≥15 years old) by history and FPG level was 9.50%, while 11.00% by history and HbA1c level. In NHES 2020, the proportions of undiagnosed patients with type 2 diabetes were 30.60% by history and FPG level, and 13.90% were untreated. Among those patients who were treated, only 47.00% had controlled FPG levels (<130 mg/dL). According to multivariable logistic regression analysis in the present study, we found that the factors significantly associated with uncontrolled diabetes included demographic data, duration of having diabetes, prescription medications, diagnosed DR, and FPG level. Most independent variables for predictive uncontrolled diabetes are nonmodifiable from the clinician's standpoint, e.g., age, gender, and type of scheme. After that, all the identified factors

**Table 2. Univariable logistic regression analysis in the development group (n = 38,568).**

| Characteristics | HbA1c | | Crude Odds ratio | 95% CI | P-value |
|---|---|---|---|---|---|
| | <9 | ≥9 | | | |
| | n (%) | n (%) | | | |
| **Gender** | | | | | |
| Male | 8,740 (34.46) | 2,289 (28.58) | 1 | | |
| Female | 16,623 (65.54) | 5,721 (71.42) | 1.31 | 1.24–1.39 | <0.001 |
| **Hospital level** | | | | | |
| Regional Hospitals | 1,930 (7.61) | 565 (7.05) | 1 | | |
| General Hospitals | 5,087 (20.06) | 1,498 (18.7) | 1.01 | 0.90–1.12 | 0.916 |
| Community Hospitals | 18,346 (72.33) | 5,947 (74.24) | 1.11 | 1.00–1.22 | 0.042 |
| **Age group (years)** | | | | | |
| <40 | 417 (1.64) | 254 (3.17) | 1 | | |
| 40–59 | 8,735 (34.44) | 3,822 (47.72) | 0.72 | 0.61–0.84 | <0.001 |
| 60–79 | 14,411 (56.82) | 3,675 (45.88) | 0.42 | 0.36–0.49 | <0.001 |
| ≥80 | 1,800 (7.10) | 259 (3.23) | 0.24 | 0.19–0.29 | <0.001 |
| **Duration of diabetes (years)** | | | | | |
| 0–9 | 13,459 (58.37) | 3,460 (47.94) | 1 | | |
| 10–19 | 8,656 (37.54) | 3,315 (45.93) | 1.49 | 1.41–1.57 | <0.001 |
| 20–29 | 838 (3.63) | 373 (5.17) | 1.73 | 1.52–1.97 | <0.001 |
| ≥30 | 105 (0.46) | 69 (0.96) | 2.56 | 1.88–3.47 | <0.001 |
| **Height (cm)** | | | | | |
| Mean ± SD | 158.03 ± 8.20 | 157.63 ± 7.97 | 0.99 | 0.99–1.00 | <0.001 |
| **Weight (kg)** | | | | | |
| Mean ± SD | 64.37 ± 13.48 | 64.35 ± 13.49 | 1 | 1.00–1.00 | 0.921 |
| **Waist (cm)** | | | | | |
| Mean ± SD | 88.85 ± 10.99 | 88.92 ± 11.14 | 1 | 1.00–1.00 | 0.618 |
| **BMI group (kg/m$^2$)** | | | | | |
| Mean ± SD | 25.72 ± 4.75 | 25.86 ± 4.81 | 1.01 | 1.00–1.01 | 0.024 |
| 18.50–22.99 | 6,262 (24.95) | 1,918 (24.18) | 1 | | |
| 23.00–24.99 | 5,008 (19.96) | 1,485 (18.72) | 0.97 | 0.90–1.05 | 0.411 |
| 25.00–29.99 | 8,790 (35.03) | 2,862 (36.08) | 1.06 | 0.99–1.14 | 0.071 |
| ≥30.00 | 4,073 (16.23) | 1,362 (17.17) | 1.09 | 1.01–1.18 | 0.031 |
| <18.50 | 961 (3.83) | 306 (3.86) | 1.04 | 0.91–1.19 | 0.583 |
| **Regions** | | | | | |
| North | 5,977 (23.57) | 1,722 (21.50) | 1 | | |
| Central | 9,096 (35.86) | 2,081 (25.98) | 0.79 | 0.74–0.85 | <0.001 |
| Northeastern | 6,002 (23.66) | 2,561 (31.97) | 1.48 | 1.38–1.59 | <0.001 |
| South | 4,288 (16.91) | 1,646 (20.55) | 1.33 | 1.23–1.44 | <0.001 |
| **Health insurance scheme** | | | | | |
| UC | 19,499 (76.88) | 6,516 (81.35) | 1 | | |
| CSMBS | 4,526 (17.84) | 1,074 (13.41) | 0.71 | 0.66–0.76 | <0.001 |
| SSS | 1,013 (3.99) | 325 (4.06) | 0.96 | 0.84–1.09 | 0.533 |
| Others | 325 (1.28) | 95 (1.19) | 0.87 | 0.69–1.10 | 0.255 |
| **Hypertension** | | | | | |
| No | 5,072 (20.00) | 2,110 (26.34) | | | |
| Yes | 20,291 (80.00) | 5,900 (73.66) | 0.70 | 0.66–0.74 | <0.001 |
| **Dyslipidemia** | | | | | |
| No | 7,255 (28.60) | 2,182 (27.24) | | | |

(*Continued*)

**Table 2.** (Continued)

| Characteristics | HbA1c | | Crude Odds ratio | 95% CI | P-value |
|---|---|---|---|---|---|
| | <9 | ≥9 | | | |
| | n (%) | n (%) | | | |
| Yes | 18,108 (71.40) | 5,828 (72.76) | 1.07 | 1.01–1.13 | 0.018 |
| **Gout** | | | | | |
| No | 23,815 (93.90) | 7,696 (96.08) | | | |
| Yes | 1,548 (6.10) | 314 (3.92) | 0.63 | 0.55–0.71 | <0.001 |
| **Renal insufficiency** | | | | | |
| No | 21,060 (83.03) | 6,771 (84.53) | | | |
| Yes | 4,303 (16.97) | 1,239 (15.47) | 0.90 | 0.84–0.96 | 0.002 |
| **Diabetic kidney diseases** | | | | | |
| No | 23,559 (92.89) | 7,417 (92.60) | | | |
| Yes | 1,804 (7.11) | 593 (7.40) | 1.04 | 0.95–1.15 | 0.38 |
| **Diabetic retinopathy** | | | | | |
| No | 24,194 (95.39) | 7,440 (92.88) | 1 | | |
| Yes | 1,168 (4.61) | 570 (7.12) | 1.59 | 1.43–1.76 | <0.001 |
| **Cerebrovascular accident** | | | | | |
| No | 24,522 (96.68) | 7,797 (97.34) | | | |
| Yes | 841 (3.32) | 213 (2.66) | 0.8 | 0.68–0.93 | 0.003 |
| **Coronary artery disease** | | | | | |
| No | 24,287 (95.76) | 7,717 (96.34) | | | |
| Yes | 1,076 (4.24) | 293 (3.66) | 0.86 | 0.75–0.98 | 0.022 |
| **Biguanide** | | | | | |
| No | 6,012 (23.70) | 1,804 (22.52) | 1 | | |
| Yes | 19,351 (76.30) | 6,206 (77.48) | 1.07 | 1.01–1.14 | 0.029 |
| **Sulfonylurea** | | | | | |
| No | 10,751 (42.39) | 2,813 (35.12) | 1 | | |
| Yes | 14,612 (57.61) | 5,197 (64.88) | 1.36 | 1.29–1.43 | <0.001 |
| **Thiazolidinediones** | | | | | |
| No | 22,391 (88.28) | 6,718 (83.87) | 1 | | |
| Yes | 2,972 (11.72) | 1,292 (16.13) | 1.45 | 1.35–1.56 | <0.001 |
| **Insulin** | | | | | |
| No | 21,580 (85.08) | 4,563 (56.97) | 1 | | |
| Yes | 3,783 (14.92) | 3,447 (43.03) | 4.31 | 4.07–4.56 | <0.001 |
| **SGLT2 Inhibitors** | | | | | |
| No | 25,304 (99.77) | 7,988 (99.73) | | | |
| Yes | 59 (0.23) | 22 (0.27) | 1.18 | 0.72–1.93 | 0.506 |
| **Calcium channel blockers** | | | | | |
| No | 14,631 (57.69) | 5,265 (65.73) | | | |
| Yes | 10,732 (42.31) | 2,745 (34.27) | 0.71 | 0.67–0.75 | <0.001 |
| **RAS blockers** | | | | | |
| No | 10,716 (42.25) | 3,672 (45.84) | | | |
| Yes | 14,647 (57.75) | 4,338 (54.16) | 0.86 | 0.82–0.91 | <0.001 |
| **Beta-blocker** | | | | | |
| No | 20,953 (82.61) | 6,884 (85.94) | | | |
| Yes | 4,410 (17.39) | 1,126 (14.06) | 0.78 | 0.72–0.83 | <0.001 |
| **Diuretics** | | | | | |
| No | 21,721 (85.64) | 7,069 (88.25) | | | |

*(Continued)*

**Table 2.** (Continued)

| Characteristics | HbA1c | | Crude Odds ratio | 95% CI | P-value |
|---|---|---|---|---|---|
| | <9 | ≥9 | | | |
| | n (%) | n (%) | | | |
| Yes | 3,641 (14.36) | 941 (11.75) | 0.79 | 0.74–0.86 | <0.001 |
| **Statins** | | | | | |
| No | 7,683 (30.29) | 2,325 (29.03) | | | |
| Yes | 17,680 (69.71) | 5,685 (70.97) | 1.06 | 1.01–1.12 | 0.031 |
| **FPG (mg/dL)** | | | | | |
| Mean ± SD | 141.76 ± 40.34 | 187.96 ± 71.50 | 1.02 | 1.02–1.02 | <0.001 |
| <100 | 2,123 (8.74) | 460 (6.04) | 1 | | |
| 100–199 | 20,429 (84.11) | 4,414 (57.93) | 1 | 0.90–1.11 | 0.958 |
| 200–299 | 1,568 (6.46) | 2,218 (29.11) | 6.53 | 5.79–7.36 | <0.001 |
| ≥300 | 167 (0.69) | 527 (6.92) | 14.56 | 11.91–17.81 | <0.001 |

SD: standard deviation, BMI: body mass index, UC: universal coverage, SSS: social security scheme, CSMBS: civil servant medical benefit scheme, FPG: fasting plasma glucose, HbA1c: hemoglobin A1c, SGLT2 inhibitors: sodium-glucose Cotransporter-2 Inhibitors, mg/dL: milligrams per deciliter, kg/m2: kilogram per square meter, kg: kilogram, cm: centimeter

were usually recorded in medical records and could be used to construct a nomogram to predict uncontrolled diabetes in routine medical services when the HbA1c test was unavailable.

For nomogram construction, related studies showed that nomograms had a better predicting prognosis than conventional methods [17–19]. A nomogram is a graphical tool that combines patients' clinical and nonclinical characteristics in a single predictive tool rather than focusing only on laboratory outcomes. The present nomogram had a concordance index of 0.77, which was far from perfect predictability. The independent variables comprised nonmodifiable and modifiable factors. The nonmodifiable factors consist of six subgroups: gender, age group, health insurance scheme, receiving hospital level, regions of Thailand, and duration of having diabetes. The modifiable factors consist of four subgroups: BMI, laboratory outcome, prescription medication, and diabetic complication.

In countries with limited resources, the prediction of uncontrolled diabetes can be challenging. Some healthcare professionals may adjust a patient's medication based solely on their fasting plasma glucose (FPG) levels. However, some patients with acceptable or equivocal FPG levels may still have uncontrolled HbA1C levels (>8% or >9%). This is called "unaware uncontrolled diabetes". In fact, it was the real risk factors for developing complications [20–22]. In addition to FPG, random postprandial glucose (RPG) testing is regarded as an effective method for evaluating short-term glycemic control and identifying individuals with uncontrolled diabetes who have normal fasting plasma glucose levels [23]. Although RPG testing may not be widely used, it could be an alternative method for physicians to provide better individualized diabetes care instead of relying solely on HbA1c levels. This method may be used in those with uncontrolled diabetes identified by this nomogram.

Then, the control method for hba1c levels in countries with limited resources should not involve monitoring a single variable but rather all variables with a simple implementation technique. The nomogram performed well, showing discrimination in the development group (AUC 0.71, 95%CI 0.70–0.72) equal to that of the validation group (AUC 0.70, 95%CI 0.70–0.71). Compared with the standard logistic regression predictive model of uncontrolled diabetes (AUC 0.69, 95%CI 0.68–0.69), the nomogram showed better accuracy than the conventional method [24]. The Ga Hyun Kim et al. nomogram predicting uncontrolled glucose

**Table 3. Multivariable logistic regression analysis for factors associated with uncontrolled diabetes in the development group (n = 38,568).**

| Characteristics | HbA1c | | Adjusted Odds ratio | 95% CI | P-value |
|---|---|---|---|---|---|
| | <9 | ≥9 | | | |
| | n (%) | n (%) | | | |
| **Gender** | | | | | |
| Male | 8,740 (34.46) | 2,289 (28.58) | 1 | | |
| Female | 16,623 (65.54) | 5,721 (71.42) | 1.25 | 1.17–1.34 | <0.001 |
| **Age group (years)** | | | | | |
| <40 | 417 (1.64) | 254 (3.17) | 1 | | |
| 40–59 | 8,735 (34.44) | 3,822 (47.72) | 0.80 | 0.65–0.97 | 0.025 |
| 60–79 | 14,411 (56.82) | 3,675 (45.88) | 0.52 | 0.42–0.63 | <0.001 |
| ≥80 | 1,800 (7.10) | 259 (3.23) | 0.35 | 0.27–0.45 | <0.001 |
| **Health insurance scheme** | | | | | |
| UC | 19,499 (76.88) | 6,516 (81.35) | 1 | | |
| CSMBS | 4,526 (17.84) | 1,074 (13.41) | 0.90 | 0.83–0.99 | 0.024 |
| SSS | 1,013 (3.99) | 325 (4.06) | 0.84 | 0.72–0.99 | 0.037 |
| Others | 325 (1.28) | 95 (1.19) | 0.78 | 0.59–1.04 | 0.088 |
| **Hospital level** | | | | | |
| Regional Hospitals | 1,930 (7.61) | 565 (7.05) | 1 | | |
| General Hospitals | 5,087 (20.06) | 1,498 (18.70) | 1.13 | 0.99–1.30 | 0.079 |
| Community Hospitals | 18,346 (72.33) | 5,947 (74.24) | 1.22 | 1.08–1.38 | 0.002 |
| **Regions** | | | | | |
| North | 5,977 (23.57) | 1,722 (21.50) | 1 | | |
| Central | 9,096 (35.86) | 2,081 (25.98) | 0.83 | 0.76–0.90 | <0.001 |
| Northeastern | 6,002 (23.66) | 2,561 (31.97) | 1.41 | 1.29–1.53 | <0.001 |
| South | 4,288 (16.91) | 1,646 (20.55) | 1.34 | 1.22–1.47 | <0.001 |
| **Duration of diabetes (years)** | | | | | |
| <10 | 13,459 (58.37) | 3,460 (47.94) | 1 | | |
| ≥10 | 9,599 (42.00) | 3,757 (52.06) | 1.24 | 1.16–1.33 | <0.001 |
| **BMI (kg/m²)** | | | | | |
| Mean ± SD | 25.72 ± 4.75 | 25.86 ± 4.81 | 0.99 | 0.98–1.00 | 0.007 |
| **FPG (mg/dL)** | | | | | |
| <100 | 2,123 (8.74) | 460 (6.04) | 1 | | |
| 100–199 | 20,429 (84.11) | 4,414 (57.93) | 1.11 | 0.99–1.25 | 0.08 |
| 200–299 | 1,568 (6.46) | 2,218 (29.11) | 5.22 | 4.56–5.98 | <0.001 |
| ≥300 | 167 (0.69) | 527 (6.92) | 9.65 | 7.7–12.09 | <0.001 |
| **Biguanide** | | | | | |
| No | 6,012 (23.7) | 1,804 (22.52) | 1 | | |
| Yes | 19,351 (76.3) | 6,206 (77.48) | 1.47 | 1.36–1.60 | <0.001 |
| **Sulfonylurea** | | | | | |
| No | 10,751 (42.39) | 2,813 (35.12) | 1 | | |
| Yes | 14,612 (57.61) | 5,197 (64.88) | 1.66 | 1.55–1.78 | <0.001 |
| **Thiazolidinediones** | | | | | |
| No | 22,391 (88.28) | 6,718 (83.87) | 1 | | |
| Yes | 2,972 (11.72) | 1,292 (16.13) | 1.36 | 1.24–1.49 | <0.001 |
| **Insulin** | | | | | |
| No | 21,580 (85.08) | 4,563 (56.97) | 1 | | |
| Yes | 3,783 (14.92) | 3,447 (43.03) | 4.31 | 3.99–4.66 | <0.001 |
| **Diabetic retinopathy** | | | | | |

*(Continued)*

**Table 3.** (Continued)

| Characteristics | HbA1c | | Adjusted Odds ratio | 95% CI | P-value |
|---|---|---|---|---|---|
| | <9 | ≥9 | | | |
| | n (%) | n (%) | | | |
| No | 24,194 (95.39) | 7,440 (92.88) | 1 | | |
| Yes | 1,168 (4.61) | 570 (7.12) | 1.19 | 1.04–1.36 | 0.01 |

SD: standard deviation, UC: universal coverage, SSS: social security scheme, CSMBS: civil servant medical benefit scheme, BMI: body mass index, FPG: fasting plasma glucose, HbA1c: hemoglobin A1c, mg/dL: milligrams per deciliter, kg/m$^2$: kilogram per square meter

found a cut-off value of 0.71. The present study included laboratory results and diabetic complications, while Ga Hyun's nomogram was constructed from history-taking factors and receiving treatment [25]. Kevin M. Pantalone et al. reported a nomogram model for uncontrolled diabetes that included 17 variables associated with the probability of HbA1c goal attainment and found that the C-index was 0.69 (95%CI 0.63–0.66) lower than that of the present study [26]. Moreover, Kevin M. Pantalone et al.'s nomogram model contained HbA1c levels and drugs that were not on the national drug list of Thailand. In addition to the methods used in the reported nomogram by Ga Hyun et al. and Kevin M. Pantalone et al., the present study performed a ROC curve and decision curve analysis to evaluate the discrimination and clinical usefulness of the method.

To support a community-based intervention, the study proposed a paradigm in which demographic information, medications, and laboratory results would be the most prevalent variables in community hospitals. Patients in a community hospital were screened annually for their HbA1c levels due to the high cost of laboratory testing and the limited budget for treating noncommunicable diseases, regardless of whether they were under control. The community hospitals were left with less money and stricter expenditure management. This model would therefore aid in screening uncontrolled patients for the suggestion of HbA1c testing, which could help community hospitals in Thailand save money, monitor patients, and prevent complications in uncontrolled diabetes patients.

The limitation our model encountered was that it aimed to make predictions rather than causal inferences conducted in a cross-sectional study. The covariates used in the model were not necessary for a causal relationship. Some covariates were medications used to improve

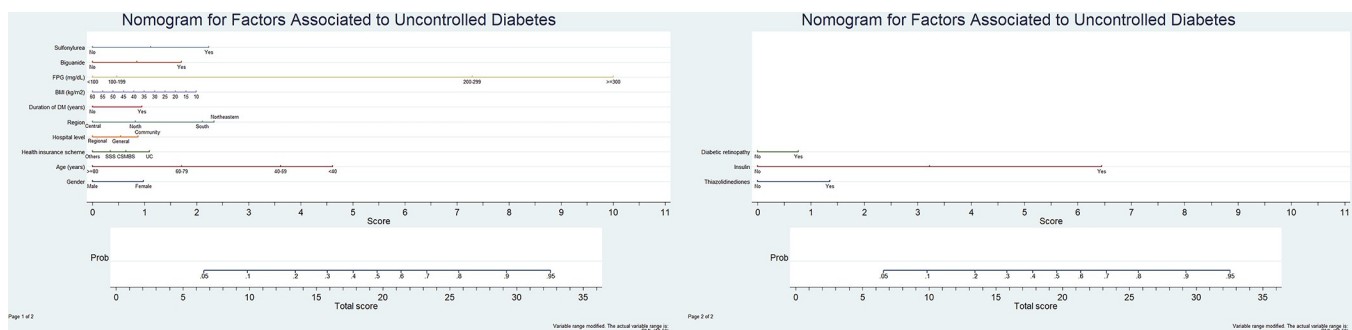

**Fig 1. Nomogram for factors associated with uncontrolled diabetes using Thailand nation-wide cross-sectional studies.** Nomogram included gender, age group, health insurance scheme, hospital level, regions of Thailand, duration of diabetes, body mass index (BMI), fasting plasma glucose (FPG) level, biguanide prescription, sulfonylurea prescription, thiazolidinedione prescription, insulin injection, diagnosis of diabetes. The nomogram is valued to obtain the probability of uncontrolled diabetes by adding up the points identified on the points scale for each variable. UC: universal coverage, SSS: social security scheme, CSMBS: civil servant medical benefit scheme, BMI: body mass index, FPG: fasting plasma glucose, HbA1c: hemoglobin A1c.

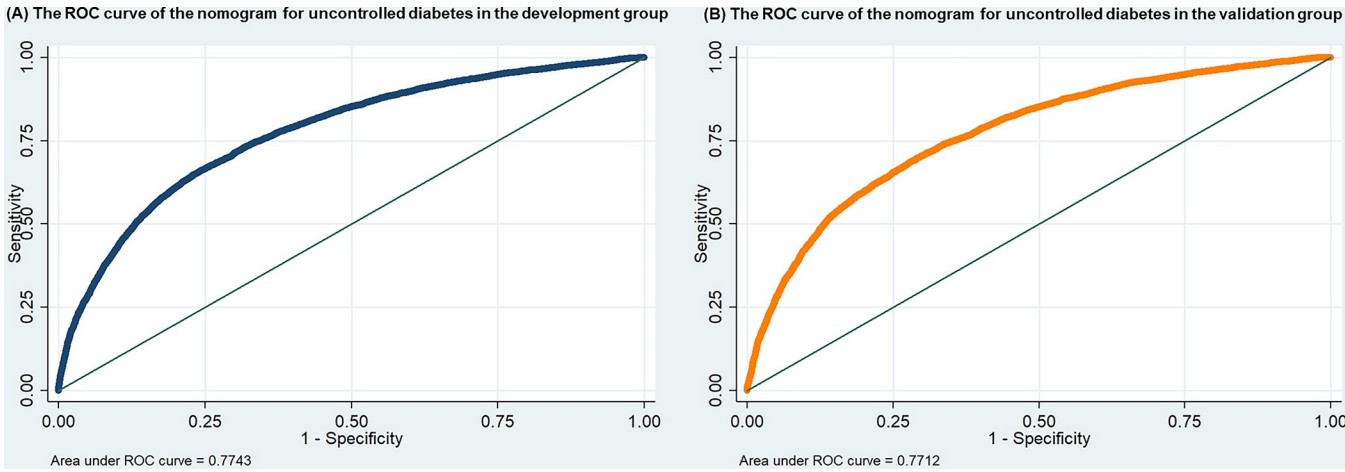

**Fig 2.** The ROC curves for uncontrolled diabetes in the development group (A) and validation group (B).

glycemic control. Incidentally, physician prescription reflects glycemic control. The fewer medications patients need to control diabetes, the lower the HbA1c must be, according to the model prediction. The full causal relationship between covariates and outcomes must be established through long-term and ongoing community-based research. Researchers strive to evaluate methods of controlling diabetes quickly and frequently. Moreover, controlled diabetes can prevent further diabetic complications such as diabetic nephropathy and diabetic retinopathy. However, dyslipidemia, which is the best predictor of cardiovascular complications, can also be evaluated by model prediction.

Further research should attempt to predict using other tools to assess uncontrolled diabetes. First, the assessment tool can create a model to analyze by region of model prediction for the whole country to analyze inaccessible drugs region. This must be one method to change the public health system. In addition, the prevention of diabetes must be developed. We already know that diabetes and metabolic syndrome are caused by insulin resistance [27]. The focus is on sugar, refined carbohydrate intake, and excessive fat consumption [28]. Therefore, the best way to prevent diabetes is to predict that one will not develop diabetes, which starts with lifestyle modification and food intake.

**Table 4. Prediction performance of the nomogram for estimating uncontrolled diabetes.**

| Prediction performance | Development | Validation |
|---|---|---|
| | Group | Group |
| **AUC (95%CI)** | 0.71 (0.70–0.72) | 0.70 (0.70–0.71) |
| **Cut-off value by Youden's index** | 0.23 | 0.23 |
| **Sensitivity, %** | 64.80 (63.60–65.90) | 69.70 (68.50–70.90) |
| **Specificity, %** | 77.00 (76.40–77.50) | 70.90 (70.20–71.60) |
| **PPV, %** | 46.70 (45.70–47.70) | 44.80 (43.70–45.80) |
| **NPV, %** | 87.50 (87.10–88.00) | 87.40 (43.70–45.80) |
| **PLR** | 2.81 (2.73–2.90) | 2.40 (2.33–2.47) |
| **NLR** | 0.46 (0.44–0.47) | 0.43 (0.41–0.44) |

AUC: Area under the curve, PPV: Positive predictive value, PPV: Negative predictive value, PLR: Positive likelihood ratio, NLR: Negative likelihood ratio

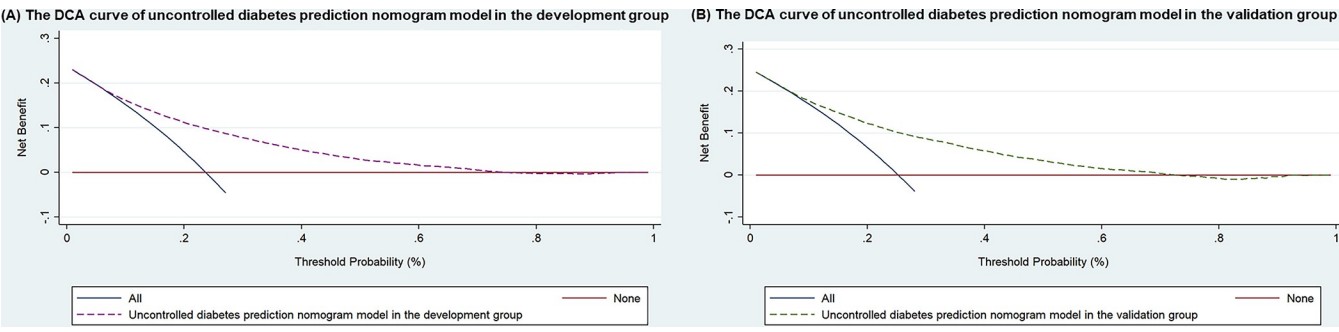

**Fig 3.** The DCA of the prediction nomogram model in the development group (A) and validation group (B).

## Conclusion

Our research demonstrated the nation-wide predictive model for uncontrolled diabetes using HbA1c ≥9% performed using baseline characteristics, medications, complications, and laboratory results. The predictive model can be used in risk adjustment of diabetes control by the prescribing physician, which should be performed more in community-based hospitals. The conducted model can identify patients at high risk of uncontrolled diabetes, which is easily used, and suitable for various purposes.

## Supporting information

**S1 Fig. Nomogram for factors associated with uncontrolled diabetes using Thailand nation-wide cross-sectional studies.** Nomogram included gender, age group, health insurance scheme, hospital level, regions of Thailand, duration of diabetes, body mass index (BMI), fasting plasma glucose (FPG) level, biguanide prescription, sulfonylurea prescription, thiazolidinedione prescription, insulin injection, diagnosis of diabetes. The nomogram is valued to obtain the probability of uncontrolled diabetes by adding up the points identified on the points scale for each variable. UC: universal coverage, SSS: social security scheme, CSMBS: civil servant medical benefit scheme, BMI: body mass index, FPG: fasting plasma glucose, HbA1c: hemoglobin A1c.
(TIF)

**S2 Fig.** The ROC curves of the nomogram for uncontrolled diabetes in the development group (A) and validation group (B).
(TIF)

**S3 Fig.** The DCA of the prediction nomogram model in the development group (A) and validation group (B).
(TIF)

**S1 Table. Comparison of demographic characteristics between the development group and validation group.** SD: standard deviation, UC: universal coverage, SSS: social security scheme, CSMBS: civil servant medical benefit scheme, FPG: fasting plasma glucose, HbA1c: hemoglobin A1c, mg/dL: milligrams per deciliter.
(DOCX)

**S2 Table. Univariable logistic regression analysis in the development group (n = 38,568).** SD: standard deviation, BMI: body mass index, UC: universal coverage, SSS: social security scheme, CSMBS: civil servant medical benefit scheme, FPG: fasting plasma glucose, HbA1c:

hemoglobin A1c, SGLT2 inhibitors: sodium-glucose Cotransporter-2 Inhibitors, mg/dL: milligrams per deciliter, kg/m2: kilogram per square meter, kg: kilogram, cm: centimeter.
(DOCX)

**S3 Table. Multivariable logistic regression analysis for factors associated with uncontrolled diabetes in the development group (n = 38,568).** SD: standard deviation, UC: universal coverage, SSS: social security scheme, CSMBS: civil servant medical benefit scheme, BMI: body mass index, FPG: fasting plasma glucose, HbA1c: hemoglobin A1c, mg/dL: milligrams per deciliter, $kg/m^2$: kilogram per square meter.
(DOCX)

**S4 Table. Prediction performance of the nomogram for estimating uncontrolled diabetes.** AUC: Area under the curve, PPV: Positive predictive value, PPV: Negative predictive value, PLR: Positive likelihood ratio, NLR: Negative likelihood ratio.
(DOCX)

## Acknowledgments

The authors would like to thank all the staff of the Department of Military and Community Medicine, Learning and Research Center for Community Medicine, Phramongkutklao College of Medicine, for their assistance with this study. The Thai DM /HT study, the Thai Medical Schools consortium medical research network (MedResNet), was supported by the Thailand Department of National Health Security Office.

## Author Contributions

**Conceptualization:** Teeraboon Lertwanichwattana, Ram Rangsin.

**Data curation:** Teeraboon Lertwanichwattana.

**Formal analysis:** Teeraboon Lertwanichwattana.

**Funding acquisition:** Teeraboon Lertwanichwattana, Ram Rangsin.

**Investigation:** Teeraboon Lertwanichwattana, Ram Rangsin.

**Methodology:** Teeraboon Lertwanichwattana, Ram Rangsin.

**Project administration:** Teeraboon Lertwanichwattana, Ram Rangsin.

**Resources:** Teeraboon Lertwanichwattana, Ram Rangsin.

**Software:** Teeraboon Lertwanichwattana.

**Supervision:** Teeraboon Lertwanichwattana, Picha Suwannahitatorn, Mathirut Mungthin.

**Validation:** Teeraboon Lertwanichwattana, Mathirut Mungthin.

**Visualization:** Teeraboon Lertwanichwattana, Mathirut Mungthin.

**Writing – original draft:** Teeraboon Lertwanichwattana.

**Writing – review & editing:** Teeraboon Lertwanichwattana, Picha Suwannahitatorn, Mathirut Mungthin, Ram Rangsin.

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
