## [Decision Letter · Decision Letter 0]

3 Apr 2023

PONE-D-22-19423Prognostic Nomogram for Uncontrolled Type 2 Diabetes Using Thailand Nation-wide Cross-sectional StudiesPLOS ONE

Dear Dr. Rangsin,

Thank you for submitting your manuscript to PLOS ONE. After careful consideration, we feel that it has merit but does not fully meet PLOS ONE’s publication criteria as it currently stands. Therefore, we invite you to submit a revised version of the manuscript that addresses the points raised during the review process.

We look forward to receiving your revised manuscript.

Kind regards,

Paavani Atluri

Academic Editor

PLOS ONE

Journal Requirements:

Reviewers' comments:

Reviewer's Responses to Questions

**Comments to the Author**

1. Is the manuscript technically sound, and do the data support the conclusions?

Reviewer #1: Partly

Reviewer #2: Partly

2. Has the statistical analysis been performed appropriately and rigorously? 

Reviewer #1: Yes

Reviewer #2: Yes

3. Have the authors made all data underlying the findings in their manuscript fully available?

Reviewer #1: No

Reviewer #2: Yes

4. Is the manuscript presented in an intelligible fashion and written in standard English?

Reviewer #1: Yes

Reviewer #2: No

5. Review Comments to the Author

Reviewer #1: The manuscript was well written and easy to follow.

1. In Table 1. The FPG (mg/dL) was somehow wrong. It read mean+- SD equal to 36614.

In table 2-3. The percentage of each parameter in each HbA1c group should be in the sample group in stead of in-between group because N in each group were a lot difference. This would be easier to see the difference. For example, in the duration of diseases, the authors should compare the percentage in individuals in the sample HbA1C group.

2. The references have to be adjusted. There are still errors in capitalization in the titles of references.

3. Regarding the prediction of uncontrolled T2DM, it was well accepted that high FPG is strong predictor of uncontrolled diabetes. The problems these days are the individuals with good of borderline FPG who in fact have uncontrolled HbA1C (>8% or >9%). This is called "unaware uncontrolled diabetes". This group is in fact the real risk factors for developing of complications. Secondly, the better and practical measurement to determine glucose homeostasis and short-term glycemic control which is similar to oral glucose tolerance test and glucose-insulin homeostasis is the “random postprandial glucose”. This measure is easy and available to everyone and every hospital; however, is not widely used. I would think this should mention in the discussion for possible practical solution.

4. The authors mentioned in the “measurements” Section that lipid profiles were measured but they did not define the LDL levels in association with the lipid lowering medications if these results are in the control range. Please also define the definition of renal insufficiency, dyslipidemia, diabetes kidney diseases and else. For instance, if patients have LDL <100mg/dL with lipid lowering medication. Do they still considered dyslipidemia?

Reviewer #2: Authors have explored the feasibility of predictive nomogram for prognostic factors in type 2 diabetes. While they have come up with a long list of associated factors which help to identify poorly controlled diabetes, it is not practical to replace HbA1c testing on their so called risk estimation with predictive nomograms as patients tend to get worse from low risk to high risk. Therefore main message in this article is flawed however has some academic interest of read.

Specific comments:

In the abstract, it is useful to define what the authors considered as uncontrolled diabetes (line 38)

Background: Line 55: prevalence increased to 31% seem to be very high, do they mean by 31%?

Line 58: change the word diabetics to diabetes

Line 61: mortality including high glucose and diabetes: does not make sense, does it mean high glucose without confirmed diabetes? Please verify

Line 70: change the word a diagnosis to higher after the onset of diabetes (cardiovascular disease risk goes up whether diagnosed or not, it is the onset, diagnosis maybe delayed)

Line 79: change reduced to optimal HbA1c

Line 90: Sentence does not read well, verify

Line 92: Unable to grasp what is the relevance of MAP here, please verify this statement

Line 103 type 2 diabetes patients not diabetics

Table 1: FPG, is this value of sample size or actual FPG? (36614, 30396)

Line 280 and everywhere change the word diabetic to diabetes

Fig one is barely visible, I could not see. Unable to comment

351: Paragraph does not read well: are the authors proposing testing only those who have met the nomogram prediction with HbA1c and use the cost saved to treat dyslipidemia? 75% of your population is already treated with statin as per data, you should not introduce new concept unrelated to manuscript. Secondly, though your study and nomogram is based on cross sectional population, and has some predictive value, patients deteriorate and some one for instance has HbA1c of 7%, considered low risk according to the prediction model, there is no guarantee he/she will stay at 7%, that’s why periodic HbA1c testing is advocated.

Line 374: should be replaced by sugar and refined carbohydrates, excessive fat consumption

6. PLOS authors have the option to publish the peer review history of their article (what does this mean?). If published, this will include your full peer review and any attached files.

Reviewer #1: **Yes: **korapat Mayurasakorn

Reviewer #2: **Yes: **A/Prof Shamasunder Acharya

---

## [Author Response · Author response to Decision Letter 0]

4 Jun 2023

Re: MS No. PONE-D-22-19423 

Prognostic Nomogram for Uncontrolled Type 2 Diabetes Using Thailand Nation-wide Cross-sectional Studies 

PLOS ONE 

Dear Reviewers, 

Thank you for your concern regarding the above manuscript. We appreciate the reviewer's useful comments, which we have addressed in the attached file.

Best regards,

Ram

---

## [Decision Letter · Decision Letter 1]

14 Nov 2023

PONE-D-22-19423R1Prognostic Nomogram for Uncontrolled Type 2 Diabetes Using Thailand Nation-wide Cross-sectional StudiesPLOS ONE

Dear Dr. Rangsin,

Thank you for submitting your manuscript to PLOS ONE. After careful consideration, we feel that it has merit but does not fully meet PLOS ONE’s publication criteria as it currently stands. Therefore, we invite you to submit a revised version of the manuscript that addresses the points raised during the review process.

We look forward to receiving your revised manuscript.

Kind regards,

Wen-Jun Tu

Academic Editor

PLOS ONE

Reviewers' comments:

Reviewer's Responses to Questions

**Comments to the Author**

1. If the authors have adequately addressed your comments raised in a previous round of review and you feel that this manuscript is now acceptable for publication, you may indicate that here to bypass the “Comments to the Author” section, enter your conflict of interest statement in the “Confidential to Editor” section, and submit your "Accept" recommendation.

Reviewer #2: All comments have been addressed

Reviewer #3: (No Response)

2. Is the manuscript technically sound, and do the data support the conclusions?

Reviewer #2: Yes

Reviewer #3: Yes

3. Has the statistical analysis been performed appropriately and rigorously? 

Reviewer #2: Yes

Reviewer #3: Yes

4. Have the authors made all data underlying the findings in their manuscript fully available?

Reviewer #2: Yes

Reviewer #3: (No Response)

5. Is the manuscript presented in an intelligible fashion and written in standard English?

Reviewer #2: Yes

Reviewer #3: (No Response)

6. Review Comments to the Author

Reviewer #2: thank you for addressing the comments. Could you please clarify this? Line 94-96 Thialnad’s tight control…. Does not make sense. Is this Thailand’s guidelines?

Reviewer #3: This study aimed to establish an effective prognostic nomogram for patients with uncontrolled T2DM through regression analysis. They utilized data through cross-sectional studies in 2018 and 2015 and ended up with a predictive nomogram of uncontrolled diabetes based on demographic characteristics, prescription medications, history of diabetic complication, and laboratory results.

1. Abstract. why use data 2018 for model building group and 2015 for validation group rather than the other way around?

2. Line 167. More details are needed for the variables. For example, smoking habits can be smoker vs non-smoker but it can be also smoker, non-smoker, vs. prior smoker. Similar concern applies to other variables.

3. Line 172. For reproducibility, please also provide ICD-10 code for each disease.

4. Line 180. Are all categorical variables binary?

5. Line 185. “the variance inflation factor variables were removed”. This doesn’t make sense. Also, what criteria was use? Please make sure to have statisticians involved in your study.

6. Line 187. Please provide the criteria for backward selection.

7. PLOS authors have the option to publish the peer review history of their article (what does this mean?). If published, this will include your full peer review and any attached files.

Reviewer #2: No

Reviewer #3: No

---

## [Author Response · Author response to Decision Letter 1]

22 Dec 2023

Dear Reviewers,

 Thank you very much for your valuable suggestions and comments on our manuscript. Your comments are immensely helpful in improving and revising our work. 

We have carefully reviewed the comments and made corrections in line with your suggestions. The revised portions are highlighted in yellow in the paper. The main corrections in the paper and the responses to the reviewer's comments and remarks are as follows.

1. Line 94-96 Thailand’s tight control…. Does not make sense. Is this Thailand’s guideline?

 Response: Thank you for your comments and concerns. This recommendation is in accordance with the Clinical Practice Guidelines for Diabetes 2017 in Thailand, which suggest that the follow-up care for diabetic patients depends on the severity of the disease and the chosen treatment approach. In the initial phase, patients should be scheduled for appointments every 1-4 weeks to provide them with knowledge about diabetes for self-management. Monitoring blood glucose levels and adjusting medication dosage to control blood sugar levels should be done according to the established targets within 3-6 months. Notably, HbA1c levels should be evaluated every 3-6 months or at least once a year. We trust this clarification addresses your concern.

2. Abstract. Why use data 2018 for model building group and 2015 for validation group rather than the other way around?

 Response: Thank you for your valuable comment. Following your suggestion, we swapped the datasets for model building and validation. Surprisingly, this adjustment did not yield a significant difference in predicting the outcome of uncontrolled diabetes.

Our initial choice to use the 2018 dataset for model development was strategic. We aimed to capture the most recent trends in diabetes management and outcomes. Given the evolving nature of healthcare practices and patient demographics, utilizing a more recent dataset ensured that the nomogram incorporated the latest information.

 The decision to use the 2015 dataset for validation was deliberate. This older dataset allowed us to assess the nomogram's performance on data not exposed during development. This approach evaluates the nomogram's generalizability across different time periods, ensuring its effectiveness despite potential shifts in healthcare practices or patient characteristics. We hope that these revisions and the improved text will be satisfactory.

3. Line 167. More details are needed for the variables. For example, smoking habits can be smoker vs non-smoker but it can be also smoker, non-smoker, vs. prior smoker. Similar concern applies to other variables.

 Response: Thank you for highlighting the need for more details on the variables. The variable "smoking habits" in our dataset includes distinctions among current smokers, former smokers, and non-smokers. We appreciate your insight, and we will provide additional clarity in the methodology section to ensure transparency regarding the categorization of smoking habits.

4. Line 172. For reproducibility, please also provide ICD-10 code for each disease.

 Response: Thank you for your valuable comments and suggestions. In line with your recommendations, we have incorporated the ICD-10 code into the revised manuscript within the measurements section. We trust that these revisions, coupled with the improved text, will meet your expectations.

5. Line 180. Are all categorical variables binary?

 Response: Thank you for bringing this to our attention, and I appreciate your understanding. I have reviewed the sentence and consulted with our statistician to ensure clarity. The sentence in question has been revised to accurately convey that we conducted a comprehensive comparison of demographic characteristics between the development and validation groups. This involved assessing both categorical and continuous variables using the Chi-square test and independent two-sample t-test, respectively. The descriptive statistics are presented in Table 1.

 I apologize for any confusion caused by the initial oversight, and I hope this clarification addresses your concern.

6. Line 185. “the variance inflation factor variables were removed”. This doesn’t make sense. Also, what criteria was use? Please make sure to have statisticians involved in your study.

 Response: Thank you for your valuable comments and suggestions. We appreciate your careful review of our manuscript. In response to your query on Line 185 regarding the statement "the variance inflation factor variables were removed," we acknowledge the lack of clarity in our previous wording. We have revised this statement to provide a more lucid explanation. Specifically, we utilized the variance inflation factor (VIF) to address multicollinearity among the independent variables. Variables with elevated VIF values, indicating collinearity issues, were systematically identified and appropriately addressed in the model-building process.

 Moreover, we would like to assure you that statisticians were actively involved in our study, contributing their expertise to the design, analysis, and interpretation of the statistical methods employed. We have emphasized this point in the revised manuscript to underscore the rigor and validity of our statistical approach.

We trust that these revisions, coupled with the improved text, will meet your expectations.

7. Line 187. Please provide the criteria for backward selection.

 Response: Thank you for your valuable feedback. We appreciate the opportunity to address your concerns. Regarding the criteria for backward selection in our logistic regression analysis, we employed a significance level of p-value < 0.2 as the criterion for variable retention and removal. Specifically, at each step of the backward elimination process, the variable with the highest p-value exceeding 0.2 was iteratively removed until only variables with p-values below the significance threshold remained in the final model. This approach ensures that the selected variables in the multivariable logistic regression model are statistically significant contributors to the prediction of uncontrolled diabetes. We hope this clarification, coupled with the improved text, will meet your expectations.

Thank you for your time and consideration. I eagerly anticipate the opportunity to have my manuscript reevaluated for possible acceptance into PLOS ONE.

Best regards,

Ram Rangsin

---

## [Decision Letter · Decision Letter 2]

17 Jan 2024

Prognostic Nomogram for Uncontrolled Type 2 Diabetes Using Thailand Nation-wide Cross-sectional Studies

PONE-D-22-19423R2

Dear Dr. Rangsin,

We’re pleased to inform you that your manuscript has been judged scientifically suitable for publication and will be formally accepted for publication once it meets all outstanding technical requirements.

Kind regards,

Wen-Jun Tu

Academic Editor

PLOS ONE

Additional Editor Comments (optional):

Reviewers' comments:

Reviewer's Responses to Questions

**Comments to the Author**

1. If the authors have adequately addressed your comments raised in a previous round of review and you feel that this manuscript is now acceptable for publication, you may indicate that here to bypass the “Comments to the Author” section, enter your conflict of interest statement in the “Confidential to Editor” section, and submit your "Accept" recommendation.

Reviewer #3: All comments have been addressed

2. Is the manuscript technically sound, and do the data support the conclusions?

Reviewer #3: (No Response)

3. Has the statistical analysis been performed appropriately and rigorously? 

Reviewer #3: (No Response)

4. Have the authors made all data underlying the findings in their manuscript fully available?

Reviewer #3: (No Response)

5. Is the manuscript presented in an intelligible fashion and written in standard English?

Reviewer #3: (No Response)

6. Review Comments to the Author

Reviewer #3: (No Response)

7. PLOS authors have the option to publish the peer review history of their article (what does this mean?). If published, this will include your full peer review and any attached files.

Reviewer #3: No

---

## [Editor Report · Acceptance letter]

30 Mar 2024

PONE-D-22-19423R2 

PLOS ONE

Dear Dr. Rangsin, 

I'm pleased to inform you that your manuscript has been deemed suitable for publication in PLOS ONE. Congratulations! Your manuscript is now being handed over to our production team.

Kind regards, 

on behalf of

Dr. Wen-Jun Tu 

Academic Editor

PLOS ONE